# The Equilibrium of Bacterial Microecosystem: Probiotics, Pathogenic Bacteria, and Natural Antimicrobial Substances in Semen

**DOI:** 10.3390/microorganisms12112253

**Published:** 2024-11-07

**Authors:** Xuelan Miao, Yanhua Zhao, Lingxi Zhu, Yutian Zeng, Cuiting Yang, Run Zhang, Arab Khan Lund, Ming Zhang

**Affiliations:** 1College of Animal Science and Technology, Sichuan Agricultural University, Chengdu 611130, China; cyanqsr@163.com (X.M.); 18482129310@139.com (Y.Z.); zhulingxi1219@163.com (L.Z.); alisaz1996@hotmail.com (Y.Z.); yangcuiting922@outlook.com (C.Y.); zhangrun202111@163.com (R.Z.); aarabbaloch@gmail.com (A.K.L.); 2Faculty of Animal Production and Technology, Shaheed Benazir Bhutto University of Veterinary and Animal Sciences, Sakrand 67210, Pakistan; 3State Key Laboratory of Swine and Poultry Breeding Industry, Sichuan Agricultural University, Chengdu 611130, China; 4Farm Animal Genetic Resources Exploration and Innovation Key Laboratory of Sichuan Province, Sichuan Agricultural University, Chengdu 611130, China

**Keywords:** semen, probiotics, pathogenic bacteria, antibacterial substances, semen quality

## Abstract

Semen is a complex fluid that contains spermatozoa and also functions as a dynamic bacterial microecosystem, comprising probiotics, pathogenic bacteria, and natural antimicrobial substances. Probiotic bacteria, such as *Lactobacillus* and *Bifidobacterium*, along with pathogenic bacteria like *Pseudomonas aeruginosa* and *Escherichia coli*, play significant roles in semen preservation and reproductive health. Studies have explored the impact of pathogenic bacteria on sperm quality, providing insights into the bacterial populations in mammalian semen and their influence on sperm function. These reviews highlight the delicate balance between beneficial and harmful bacteria, alongside the role of natural antimicrobial substances that help maintain this equilibrium. Moreover, we discuss the presence and roles of antimicrobial substances in semen, such as lysozyme, secretory leukocyte peptidase inhibitors, lactoferrin, and antimicrobial peptides, as well as emerging antibacterial substances like amyloid proteins. Understanding the interactions among probiotics, pathogens, and antimicrobial agents is crucial for elucidating semen preservation and fertility mechanisms. Additionally, the potential for adding probiotic bacteria with recombinant antibacterial properties presents a promising avenue for the development of new semen extenders. This review offers updated insights to understand the equilibrium of the bacterial microecosystem in semen and points toward innovative approaches for improving semen preservation.

## 1. Introduction

Semen contains a large amount of nutrients, making it easy for bacteria to grow. According to whether bacteria are beneficial or harmful to animals, these bacteria present in semen usually are classified into probiotics and pathogens. Probiotics play a beneficial role in semen preservation and maintenance of semen quality. Conversely, pathogenic bacteria have the potential to adversely affect semen quality and reproductive performance. Studies and reviews [1,2,3,4,5,6,7,8,9,10,11,12,13] mainly focus on the impact of pathogenic bacteria on sperm quality, as well as the effect of altering seminal plasma metabolites on sperm quality. However, it is noteworthy that mammalian semen contains natural antibacterial substances that exert antibacterial effects and play a crucial role in the equilibrium between probiotics and pathogenic bacteria. The interaction among probiotics, pathogenic bacteria, and antibacterial substances establishes a bacterial microecosystem in semen for sperm survival [14,15,16]. In this review, we aim to summarize it systematically and focus on the equilibrium between pathogenic bacteria, probiotics, and antimicrobial substances.

In the current global context, declining semen quality and increasing antibiotic resistance pose a dual threat to semen quality and reproductive health in mammals, and this phenomenon challenges mammalian reproductive efficiency. Therefore, a deeper understanding is crucial about how probiotics and natural antimicrobial substances in semen counteract the deleterious effects of pathogenic bacteria, and how to maintain the flora equilibrium in semen. That involves studies on the mechanisms of interactions between bacteria and antimicrobial substances, as well as exploring antibiotic alternatives, such as natural antimicrobial substances, in semen preservation. The knowledge not only contributes to our better understanding of the biological defense mechanisms of semen but also provides potential intervention strategies for semen storage and reproductive health, which can effectively improve semen preservation quality and fertility, providing long-term benefits for mammalian reproductive health.

## 2. Bacteria in Semen

### 2.1. Sources of Bacteria in Semen

Bacteria may arise from localized or generalized infections within the mammalian body. In addition, they may also come from exogenous contamination such as feces, respiratory secretions, skin, hair, and the various stages of semen processing. Currently, there are strict hygienic standards and sterilization requirements for all aspects of the breeding farm to minimize the possibility of exogenous contamination of bacteria, so we focus on the interpretation of endogenous bacteria in semen. Endogenous bacteria in semen may originate in the upper reproductive tract, hematogenous or lymphogenous dissemination of bacteria from the intestinal or oral microbiota [17], or bacterial circulation through the bloodstream [18].

Semen makes up 90% of the volume of ejaculated semen and consists of secretions from the epididymis, prostate, seminal vesicles, urethral bulbourethral glands, and periurethral glands [19,20]. It has been found that semen from different parts of the reproductive tract contains the same dominant bacteria, but the abundance of the various dominant bacteria varies slightly, possibly due to the fact that different parts of the reproductive tract contain slightly different abundances of the various bacteria. The microbiota of male testicular samples [21], seminal vesicle samples [22], and prostate samples [23] were analyzed separately, and it was found that all three had the same dominant bacteria: *Lactobacillus*, *Pseudomonas*, *Escherichia coli*, and *Staphylococcus aureus*, but the dominant bacteria had slightly different abundances in different samples. Through these studies, it was found that these dominant bacteria were also the dominant bacteria in semen, and their abundance in semen is shown in Figure 1 [24,25,26,27].

### 2.2. Types of Bacteria in Semen

The current investigation [28,29,30,31,32] reveals a substantial diversity and abundance of bacterial species in the semen of domestic animals. Although there are slight differences in the different species, the bacterial types and dominant bacteria are similar. A series of bacteria such as *Escherichia coli*, *Staphylococcus*, *Pseudomonas aeruginosa*, *Lactobacillus*, *Ureaplasma urealyticum*, *Clostridium trachomatis*, *Gonococcus*, and *Streptococcus pyogenes* have already been found in mammalian semen from boars [7], bovines [4], rabbits [29], mice [30], and horse [25]. In particular, *Escherichia coli*, *Pseudomonas aeruginosa*, *Lactobacillus*, *Staphylococcus*, and *Streptococcus* were more abundant in these samples, which was consistent with our findings. We identified 2783 bacterial species in 226 semen samples from nine breed boars (Figure 2A). Predominant bacteria include *Escherichia coli*, *Pseudomonas aeruginosa*, *Lactobacillus*, *Staphylococcus*, and *Streptococcus*, constituting a noteworthy proportion of the microbial population. We also found the abundance of probiotics and pathogenic bacteria have a significant correlation in boar semen (Figure 2B), which suggests the intrinsic interaction between probiotics and pathogenic bacteria.

### 2.3. Effects of Bacteria on Semen Quality

Bacteria play a pivotal role in semen, significantly impacting key sperm indicators including motility [32], viability [33], plasma membrane integrity [33,34], and acrosome integrity [33,34]. Semen contaminated with bacteria can result in decreased quality, characterized by a shortened shelf life and a severe decline in sperm motility [33,34]. Additionally, bacteria in semen can cause reproductive tract infections in females, which can lead to early embryo implantation failures, fetal deaths, abortions, etc., resulting in huge economic losses [28,35]. The mechanisms [33,35,36] through which bacteria affect the duration and efficacy of semen preservation involve nutrient competition, alterations in the dilution environment, osmotic pressure leading to acrosome swelling, inclusion of agglutination factors causing sperm agglutination, and acid production from anaerobic fermentation, resulting in a pH decrease to 5.7–6.4. It is noteworthy that mammalian semen typically harbors both probiotic and pathogenic bacteria, and their influences on sperm are distinct. Common probiotics and pathogens are outlined in Table 1.

#### 2.3.1. Effects of Probiotics on Semen Quality

Probiotics, living microorganisms, exert various effects on host health, such as inhibiting pathogen growth [45,46], enhancing intestinal barrier function [46,47,48], modulating the immune system [49], and influencing pain perception. These mechanisms contribute to improved host health, growth performance, disease resistance, intestinal health, and reproductive system function [50,51]. In particular, the role of probiotics in protecting spermatozoa is crucial for improving the reproductive performance of animals [52]. Probiotics are not only positively correlated with sperm viability parameters, structural integrity, and sperm capacitation, but also have antagonistic effects on pathogenic bacteria (Figure 3).

Probiotics in semen include *Lactobacillus*, *Bifidobacterium*, *Lactobacillus rhamnosus,* and *Bacillus subtilis*. Among these, *Bifidobacteria* (Gram-positive bacteria, belonging to the phylum Actinobacteria) and *Lactobacilli* (Gram-positive bacteria, belonging to the phylum Firmicutes) are the two most abundant probiotics with the highest abundance in the semen of common livestock. Their abundance in seminal fluid is shown in Figure 1. Probiotics in semen can improve sperm viability and contribute to the quality of semen preservation. Although semen quality is influenced by multiple factors like environment, lifestyle habits, nutritional status, and genetics, it has been shown that *Lactobacillus* and *Bifidobacterium* act as antioxidant supplements to reduce intracellular hydrogen peroxide levels in sperm, reduce DNA breaks, and improve sperm viability. The abundance of the two was found to be significantly positively correlated with the quality of spermatozoa and fertility [25,38]. Therefore, with full consideration of the effects of other factors, it is also possible to help improve semen quality by adding probiotics. Additionally, semen quality was found to be seasonally related by comparing semen from different seasons [25]. Sperm quality and fertility are better in winter when probiotics dominate, especially *Lactobacillus* and *Bifidobacterium*, and vice versa in summer [25]. The difference in sperm quality between winter and summer suggests that seasonal changes may indirectly affect fertility by affecting the microbiological equilibrium in the body. Future studies should further explore the relationship between probiotics and sperm quality, and consider how to apply these findings to clinical practice in order to improve fertility rates.

#### 2.3.2. Effects of Pathogenic Bacteria on Semen Quality

Pathogenic bacteria are bacteria that can cause disease in the organism, having detrimental effects on sperm viability parameters, structural integrity, and sperm capacitation with the potential to cause poor reproductive performance [40,53]. Common pathogenic bacteria are shown in Table 1.

*Pseudomonas aeruginosa*, *Escherichia coli,* and *Prevotella* are three Gram-negative bacteria, the first two belonging to the phylum Proteobacteria, and *Prevotella* belonging to the phylum Bacteroidetes. As three dominant pathogenic bacteria commonly found in boar semen, they have significant negative effects on spermatozoa: *Pseudomonas aeruginosa* disrupts sperm acrosomes [35,36], plasma membrane [35,36], and mitochondria [2]. *Escherichia coli* causes sperm agglutination [33,45]. *Prevotella* is negatively correlated with sperm concentration, viability, and morphology [8]. Further studies have shown that the presence of these pathogenic bacteria is directly related to sperm quality and reproductive potential. *Pseudomonas aeruginosa* can affect sperm motility by interfering with energy metabolism and post-translational modification of proteins [12,13], and high abundance of *Pseudomonas aeruginosa* in semen samples is negatively correlated with reduced sperm quality and reproductive potential [25,42]. *Escherichia coli*, then, affects sperm quality through its surface structure and soluble factors, like bacterial lipopolysaccharide (LPS) [54,55,56]. LPS has been shown to reduce sperm quality in mammals, and *Escherichia coli*’s soluble factors may negatively affect spermatozoa [45,50,57]. Additionally, the hair of type 1 and type P *Escherichia coli* may affect sperm mitochondrial function [51], and mannose on the sperm surface plays a key role in this interaction. *Prevotella* is associated with sperm viability and morphological defects, suggesting that it may play a potential role in infertility [58,59].

*Staphylococcus aureus* is a Gram-positive bacteria belonging to the phylum Firmicutes, and infection with *Staphylococcus aureus* causes reproductive disorders and greatly reduces sperm viability in male mammals [56,57,58]. It was found that *Staphylococcus aureus* works through the interference of energy metabolism processes in spermatozoa [59,60,61,62]. Specifically, *Staphylococcus aureus* reduces the activity of glyceraldehyde-3-phosphate dehydrogenase (GAPDH) during glucose metabolism, thereby inhibiting ATP production. The lack of energy supply ultimately leads to decreased sperm motility, as well as other sperm functions. Thus, the presence of *Staphylococcus aureus* may have a significant negative impact on the overall sperm quality and fertility.

Pathogenic bacteria in mammalian semen may pose a threat to male reproductive health and affect female reproductive health via sexual transmission and artificial insemination. Additionally, by the fact that increased bacterial contamination in semen can lead to a reduction in semen quality [40,53], we can further infer that pathogenic bacteria can adversely affect artificial insemination efficiency and semen exchange, which may become an important limiting factor in the implementation of co-breeding and breeding improvement. Therefore, studies on the prevention and control of pathogenic bacteria are of great practical importance.

## 3. Natural Antimicrobial Substances in Semen

Semen contains various natural antimicrobial substances. Natural antimicrobial substances are a type of substance that inhibits the harmful effects of pathogenic bacteria in semen and exerts an antimicrobial effect. Some of these substances adhere to the spermatozoa membrane, while others are dissolved in seminal plasma. They exert their antimicrobial effects through direct or indirect actions, resulting in the inhibition and elimination of bacteria. This antimicrobial activity of natural antimicrobial substances in semen has been documented across various species and demonstrates efficacy against a broad spectrum of bacterial species, such as *Escherichia coli*, *Staphylococcus aureus*, *Pseudomonas aeruginosa*, *Enterobacter aerogenes*, and *Enterobacter cloacae* [54,55,56,57,60,61]. This antimicrobial activity exhibited in semen through natural antimicrobial substances is essential for the elimination of pathogenic bacteria, protection of spermatozoa, and maintenance of reproductive health [32].

In recent years, researchers have identified several substances that could elucidate the antimicrobial activity of semen [62], such as Lysozyme (LSZ), which may possess bactericidal properties in various mammalian secretions such as semen, mucus, and saliva. Moreover, antimicrobial assays and gel electrophoresis of antimicrobial substances extracted from male semen have confirmed the presence of several antimicrobial peptides, such as secretory leukocyte peptidase inhibitor (SLPI), group II phospholipase A2 (PLA2), lactoferrin (LF), and human cationic antimicrobial peptide-18 (hCAP-18). These peptides exhibit antimicrobial activity against a broad spectrum of bacteria, including Streptococcus and Gonococcus [5,53]. The common natural antimicrobial substances in semen and their mechanisms of action are detailed in Table 2.

### 3.1. Lysozyme (LSZ)

LSZ, also known as muramidase, is an alkaline hydrolase and a non-specific immunoprotein. LSZ is widely distributed in phages, bacteria, plants, and animals [74,75], including the semen of pigs [76,77], cattle [78], and horses [79], where it plays a vital role in the defense of the body against microorganisms [63]. LSZ is able to hydrolyze peptidoglycan in the cell wall of pathogenic bacteria (Figure 4), which is a component of the cell wall in Gram-positive bacteria. LSZ hydrolyzes the β-1,4 glycosidic bond in N-acetyl cytosolic acid and N-acetyl glucosamine in peptidoglycan, leading to rupture of the Gram-positive bacterial cell wall under osmotic pressure, resulting in lysis [80]. Additionally, some LSZ can induce bacterial lysis by stimulating autolysin activity upon interaction with the cell surface [63,64].

Based on their origin and structural properties, LSZs can be classified into several types: type C LSZ, type G LSZ, type I LSZ, phage LSZ, and plant LSZ [63,64]. In mammals, type C LSZ is particularly common, and it mainly acts against pathogenic microorganisms [81]. In the field of reproductive health, LSZ shows particular importance. Studies have demonstrated that LSZ is present in mammalian semen and has significant antimicrobial activity against pathogenic bacteria like *Escherichia coli* and *Staphylococcus aureus* [76,77,78,79], which contributes to the enhancement of sperm quality. Therefore, the presence of LSZ and its activity levels have the potential to serve as biomarkers for assessing fertility and sperm health.

### 3.2. Secretory Leukocyte Peptidase Inhibitor (SLPI)

SLPI, also known as anti-leukocyte peptidases or secretory peptidase inhibitors, is a multifunctional biomolecule that is widely present in body fluids like semen and saliva. The carboxy-terminal functional domain of SLPI exhibits broad-spectrum antimicrobial activity [62,66,67], indicating that it plays an important role in biological defense mechanisms. Additionally, SLPI has anti-inflammatory and tissue repair-promoting effects [79,80]. The action mechanism of SLPI is related to the special structure of the peptide chain, but the specifics need to be explored further.

Studies have confirmed that SLPI inhibits the effects of harmful bacteria on spermatozoa and restores sperm viability in a dose-dependent manner, underscoring the potential importance of SLPI in protecting spermatozoa and maintaining reproductive health [62]. In short, SLPI, as a neutrophil elastase inhibitor present in body fluids like semen, not only plays a role in antimicrobial activity, but also in promoting wound healing and protecting spermatozoa, and these multiple functions make SLPI a molecule worthy of intensive study in the biomedical field. Meanwhile, exploring the variation of SLPI concentration in semen and its relationship with reproductive health indicators may help to reveal its broader role in systemic defense, thus developing new diagnostic tools and therapeutic approaches.

### 3.3. Lactoferrin (LF)

LF is a glycoprotein involved in iron transport and storage, found in various secretions of the body (like semen and saliva) as well as in the mucosal epithelium and neutrophils of the gastric, colon, lung, and reproductive tract [68,69]. LF possesses antioxidant and broad-spectrum antimicrobial capabilities [82,83], and it binds to iron tightly so that the bacteria are unable to obtain the necessary iron for their growth, thus inhibiting and killing the bacteria.

The antimicrobial properties of LF show promise in influencing sperm functional parameters and enhancing in vitro fertilization [84], which not only reduces the deleterious effects of harmful bacteria on spermatozoa but also regulates various aspects of the reproductive process [84,85]. Additionally, LF also enhances LSZ activity to exert antimicrobial effects, thus protecting spermatozoa [69]. The study of LF is an exciting area, however we still need more research to fully understand the mechanisms and potential side effects.

### 3.4. Antibacterial Peptides (AMPs)

AMPs are a group of alkaline endogenous peptides that are present in various secretions of the body (like semen and saliva) as well as in the skin [86], the digestive tract [87], the respiratory system [88], and the reproductive tract [89,90], with a broad spectrum of antimicrobial activity, effectively eliminating target pathogens [91,92]. Beyond their antimicrobial role, AMPs also contribute to cell proliferation [93], wound healing [94], angiogenesis [95], and the response to acute inflammation [96] (Figure 5). The amphiphilic structure [60] of AMPs and the spatial separation of their cationic and hydrophobic components are essential for their effective interaction with bacterial membranes [70]. This structural feature enables AMPs to interact with lipids in asymmetric bacterial membranes similarly [71]. Electrostatic interactions facilitate peptide binding to negatively charged bacterial lipid head groups such as phosphatidylglycerol and cardiolipin [70], while hydrophobic interactions allow peptides to penetrate the lipid bilayer’s hydrophobic regions, destabilizing bacterial cell membranes compared to conventional antibiotic [60].

It was found that the exposure of anionic sulfo-galactosyl-glycero-lipid (SGG) and the relatively low cholesterol content in porcine spermatozoa membranes [91,92] render sperm cells potentially susceptible to AMPs [93], resulting in a sperm-protective effect. Additionally, although most AMPs can directly kill various microbial pathogens such as bacteria, yeasts, fungi, and viruses, and modulate host immunity [94,95,96], many have a limited spectrum of activity and are effective only at high concentrations, which can increase their cytotoxicity [86,97]. These insights provide directions for future studies on how to develop safer and more effective antimicrobial strategies by improving the structure of AMPs or discovering new regulatory mechanisms to enhance their antimicrobial effects while reducing potential toxicity to host cells.

Currently, all AMPs have been shown to have antimicrobial activity, but only a few have antiviral activity [87]. Defensins exhibit roles in antiviral immunity [88,89], which may exert their antiviral activity by altering the innate immune response induced by viral infection [88], and may also block viral infection by acting directly on viral particles or by indirectly intervening at various stages of the viral life cycle [89]. Additionally, the antimicrobial peptide LL-37 exerts its antiviral activity by interacting directly with the envelope and protein capsid [90,98]. Although the current use of AMPs as antiviral therapy has great appeal and some successful in vitro results, the widespread use of AMPs as antiviral therapy still requires further research.

### 3.5. Group II Phospholipase A2 (PLA2)

PLA2 enzyme, as a phospholipid Sn-2 lipase, is found in various body fluids, including blood, tears, and seminal fluid. PLA2 is involved in various biological processes like cell signaling, inflammatory response, and immune regulation [99]. Additionally, PLA2 can exert its antimicrobial effect by hydrolyzing phospholipids on the cell membranes of certain Gram-positive bacteria, causing the bacteria to rupture and die, as well as by activating the body’s immune system, which kills a number of Gram-negative bacteria with the help of complements and other factors [72,73].

The elevated levels of PLA2 in semen could potentially serve a crucial role in safeguarding the sperm surface against bacterial threats. Bovine seminal fluid contains both calcium-dependent and calcium-independent PLA2 [100,101]; they both show a clear affinity for the sperm surface and can exert an antimicrobial effect, exerting a protective effect on the spermatozoa and thus maintaining reproductive health. In mammalian semen, the presence of PLA2 not only correlates with the energization and maturation of spermatozoa but also directly participates in the acrosome reaction, a process vital for male fertility [72,73]. The role of PLA2 in semen quality and reproductive health cannot be ignored, but further studies are still needed to reveal more details about PLA2 in sperm protection and reproductive health and to provide new strategies for improving fertility and treating related diseases.

### 3.6. Others

Zinc ions (Zn^2+^), semenogelin (SG), SGI-derived peptides, and HEL-75 protein are also found in semen, exhibiting varying degrees of antimicrobial activity. Zn^2+^ is a crucial metal ion in organisms, with mammalian prostate fluid containing a high concentration of it. Following sperm emission, prostate fluid mixes with semen coagulation protein (SG) secreted by seminal vesicle glands. SG competes for Zn^2+^ binding and activates prostate-specific antigen (PSA). PSA activation leads to the degradation of SG, causing semen liquefaction and the release of SG-degrading peptides. These peptide fragments possess diverse levels of antimicrobial activity and can protect sperm from the negative effects of bacteria, thereby maintaining reproductive tract health [53].

Usually, antibiotics, such as penicillin, streptomycin, gentamicin, or their mixture are added to semen extenders to mitigate bacterial contamination [33,36,102]. However, recent years have seen an increase in bacterial resistance to antibiotics, and the requirement for antibiotic-free production. The emergence of natural antimicrobial substances has become an advantageous option. Natural antimicrobial substances in semen play an important role in the body’s immune defense, and research on adding natural antimicrobial substances to semen has focused on exploring their effects on the male reproductive system, including semen quality, sperm function, and potential therapeutic effects on reproductive tract infections. It has been found that LL-37 [103], defensins [104], cathelicidins [105] and histatins [106], as antimicrobial peptides naturally occurring in the organism, have a protective effect on spermatozoa when they are added to semen. This can improve the antimicrobial capacity of semen, reduce reproductive tract infections, and can effectively solve the problem of antibiotic use due to drug resistance problem. Currently, natural antimicrobial substances still need more profound research to make them a powerful measure to solve the bacterial contamination of semen.

## 4. Interaction of Bacterial Microecosystem: Probiotics, Pathogenic Bacteria, and Natural Antibacterial Substances

### 4.1. The Relationship Between Probiotics and Pathogenic Bacteria

The probiotics and pathogenic bacteria in semen work together to maintain a flora equilibrium through interaction and competition, which is essential for sperm function and health. Firstly, probiotics like *Lactobacillus* can inhibit the growth of pathogenic bacteria like *E. coli* [14,39], *Pseudomonas aeruginosa* [40], *Prevotella* [42], and *Haemophilus* [7] through the production of antimicrobial substances, thus protecting sperms from their adverse effects. Secondly, probiotics also provide essential nutritional support to sperm to promote their survival and vitality [16,50]. These probiotics not only help maintain the health of the reproductive tract but also provide a safer environment for sperm to survive. However, when pathogenic bacteria increase in number and upset the flora equilibrium, they can cause reproductive tract infections, impair sperm function, and even lead to infertility [40,53]. The presence of these pathogenic bacteria not only increases the risk of infection but also negatively impacts the environment in which sperm can live.

Therefore, maintaining equilibrium between probiotic and pathogenic bacteria in semen is essential to protect sperm, reduce the risk of infection, and preserve reproductive health. Maintaining this equilibrium requires an intensive understanding of the interaction mechanisms between probiotics and pathogenic bacteria, as well as how to modulate the microbial community in semen through lifestyle, diet, and probiotic supplementation.

### 4.2. The Equilibrium of Probiotics, Pathogenic Bacteria, and Natural Antibacterial Substances

The bacterial microecosystem in semen is in a complex equilibrium of probiotics, pathogenic bacteria, and natural antimicrobial substances. This equilibrium is critical for sperm survival, preservation, and the insemination process, affecting the health and fertility of the spermatozoa.

Immunological, reproductive, genetic, and endocrine factors significantly influence the equilibrium of the bacterial microecosystem [14,15,16], regulating the microbial community in semen through different mechanisms (Figure 6). Probiotics exhibit antimicrobial properties by releasing metabolites with properties like antioxidant, anti-inflammatory, pH-regulating, and energy-influencing that directly hinder the growth and proliferation of pathogenic bacteria [107]. At the same time, cytokines produced by immune and non-immune cells within the body have an impact on sperm quality and quantity. For example, pro-inflammatory cytokines like interleukin (IL)-1, IL-6, IL-8, and tumor necrosis factor (TNF)-α may adversely affect spermatogenesis and sperm function [108], and transforming growth factor (TGF)-β is involved in the regulation of cell proliferation and differentiation and influences spermatogenesis and maturation [109]. Additionally, cytokines also modulate natural antimicrobial substances so that they can effectively exert antimicrobial and bactericidal roles, and inhibit the harmful effects of pathogenic bacteria along with probiotics [110,111]. Nutrients, like proteins [112], amino acids [112], lipids [113], vitamins [114], and minerals [115], also play a role in maintaining the dynamic equilibrium among probiotics, pathogenic bacteria, and natural antimicrobial substances.

Currently, the specific mechanisms and influencing factors of bacterial equilibrium in semen are yet to be further studied and elucidated. Future studies could focus on the interactions between bacterial equilibrium in semen and microecosystems in other parts of the body, to comprehensively understand the impact of bacterial microecosystem equilibrium in semen on sperm health, and to provide a scientific basis and effective strategies for semen preservation and fertility enhancement.

## 5. Applicability and Prospects

Current research in reproductive health has focused on different aspects. On the one hand, research has focused on the impact of pathogenic bacteria on sperm quality and metabolites in semen, while also examining the efficacy of natural antimicrobials against pathogenic effects. The studies have clearly outlined the pathways through which pathogenic bacteria affect reproductive health, providing a deeper and more comprehensive understanding of the linkages. However, the role of probiotics in semen protection is relatively absent in the reproductive health landscape, but there are promising research directions and application strategies. For example, probiotics can be used to express natural antimicrobial substances to effectively inhibit pathogenic bacteria, and the use of lysogenic metabolites of pathogenic bacteria to trigger the expression of natural antimicrobial substances in probiotics may be a promising means of maintaining the semen microecological equilibrium (Figure 7). In addition, semen is deeply linked to mammalian reproduction and health. Based on this close relationship, it is possible to develop and fully utilize the advantages of probiotics and various natural antimicrobial substances by further studying the composition and characteristics of the semen microbiome, and then combining them with mature artificial insemination techniques. Through comprehensive research and practical application, it is expected that a series of new strategies can be successfully developed to improve semen quality and effectively enhance the reproductive capacity of animals, thus injecting new vitality and development momentum into the field of animal reproductive health.

In upcoming studies, attention can be directed towards understanding the role and mechanism of amyloid proteins in semen regarding antimicrobial properties. Amyloid, found in normal semen of healthy, young men, contributes significantly to the semen’s ability to enhance HIV infection. However, emerging evidence suggests that amyloid in semen may confer evolutionary advantages in terms of survival or reproductive success. Functional amyloid likely plays crucial roles in various reproductive processes such as gametogenesis and fertilization. Therefore, comprehensive investigations into the role and mechanism of action of amyloid in semen are urgently required. Additionally, the coden DNA region of natural antimicrobial substances insert these probiotic bacteria to enhance the antimicrobial capacity of semen, raise the quality of sperm and fertilization rate, and improve reproductive health through new technologies such as gene editing also urgently needs further research.

## Figures and Tables

**Figure 1 microorganisms-12-02253-f001:**
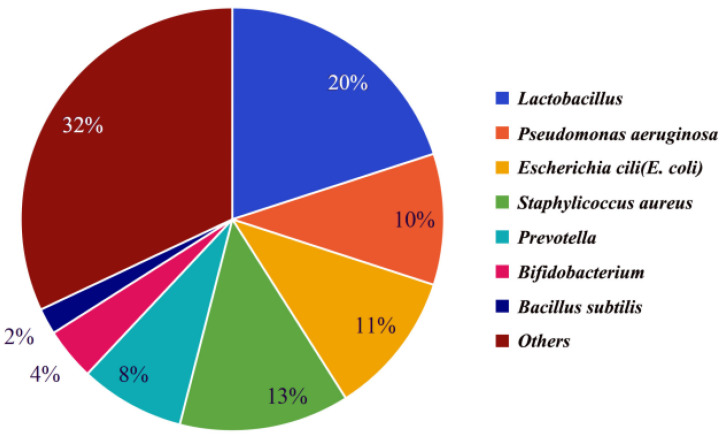
The abundance of bacteria in seminal fluid of common livestock. The percentage indicates the proportion of top-7 bacteria to the total bacteria.

**Figure 2 microorganisms-12-02253-f002:**
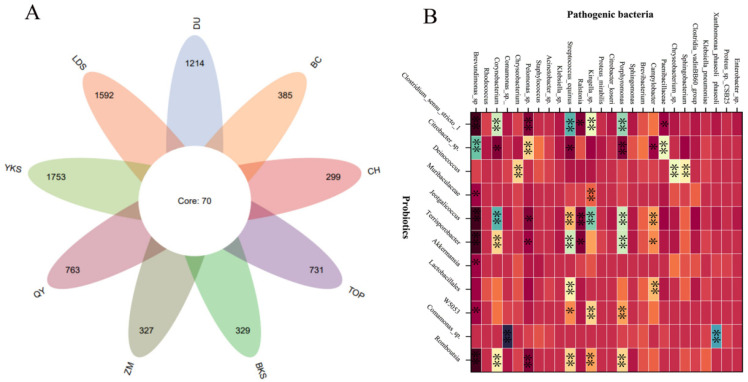
The bacterial species and abundance correlation in boar semen. (**A**) A Venn diagram of 2783 bacterial species from 9 breed boars. (**B**) The abundant correlation between top-11 probiotics and top-26 pathogenic bacteria. YKS: Yorkshire, LDS: Landrace, DU: Duroc, BKS: Berkshire, TOP: Topek boar, ZM: Zangmei, QY: Qingyu boar, CH: Chenghua boar, BC: Bacheng boar. * significant correlation (*p* < 0.05), ** extremely significant correlation (*p* < 0.01).

**Figure 3 microorganisms-12-02253-f003:**
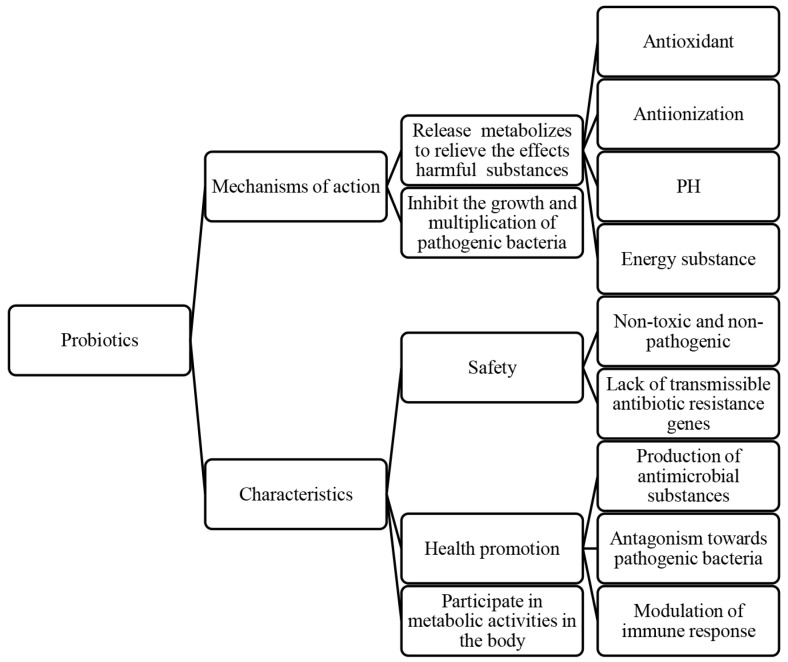
The protective effects of probiotics on semen and their characteristics.

**Figure 4 microorganisms-12-02253-f004:**
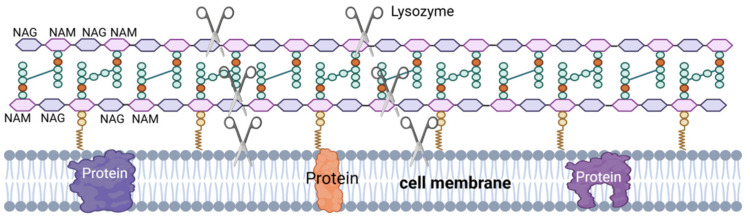
LSZ’s mechanism of action. NAG–NAM: LSZ hydrolyzes the β-1,4 glycosidic bond between the N-acetylmuramic acid (NAM) monomer and the adjacent N-acetyglucosamine (NAG) monomer. Hydrolysis of PG by LSZ leads to cell wall instability and bacterial cell death. III: LSZ can also have a bactericidal effect through the mechanism of its cationic nature, the formation of pores in the negatively charged bacterial cell membranes by lysozyme.

**Figure 5 microorganisms-12-02253-f005:**
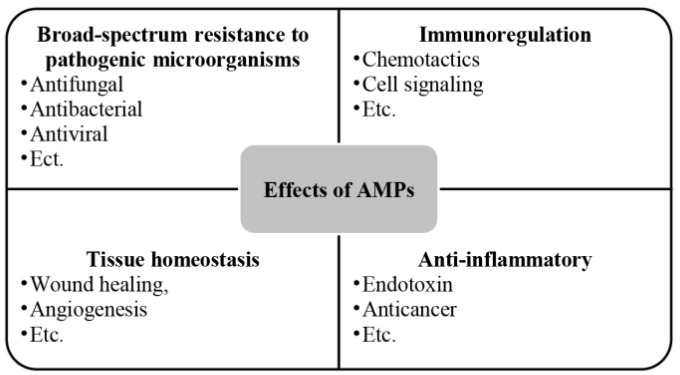
Effects of AMPs.

**Figure 6 microorganisms-12-02253-f006:**
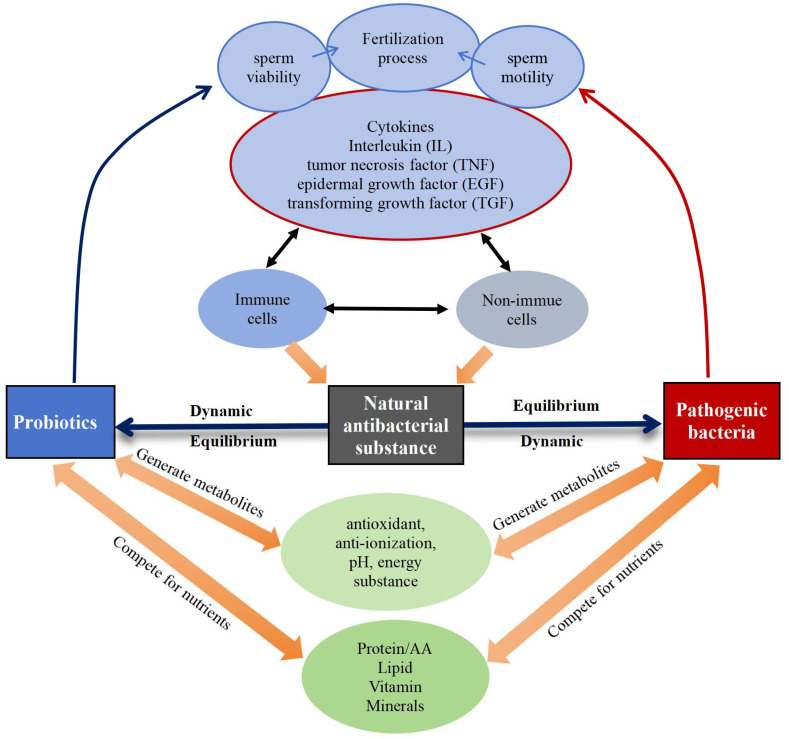
Dynamic equilibrium of probiotics, pathogenic bacteria, and natural antimicrobial substances.

**Figure 7 microorganisms-12-02253-f007:**
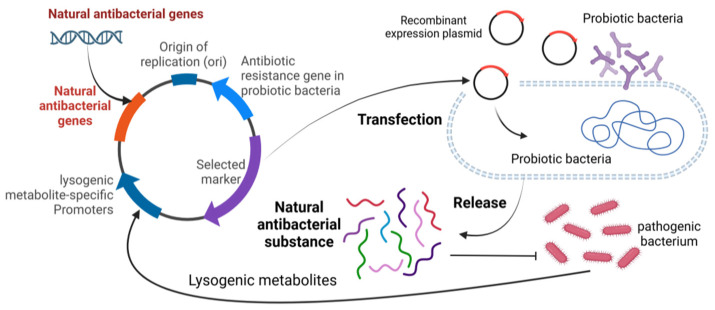
Schematic diagram of using probiotics to produce natural antibacterial substances.

**Table 1 microorganisms-12-02253-t001:** Common probiotics and pathogens in boar semen.

Type	Bacterium	Effects on Sperm Quality	References
probiotics	*Lactobacillus*	·Positively correlate with sperm viability parameters, structural integrity, and capacitation ·Have antagonistic effect with pathogenic bacteria	[37,38,39]
	*Bifidobacterium*	·Improve sperm motility ·Reduce DNA fragmentation ·Reduces intracellular oxidative stress	[40]
	*Lactobacillus rhamnosus*	·Used in reproduction, oocyte maturation ·Supplements to improve spermatogenesis ·Enhance sperm kinematic parameters	[41,42]
	*Lactobacillus paracasei*	·Reduce intracellular oxidative stress ·Stop DNA breaks ·Reduce sperm DNA loss	[42]
	*Bacillus subtilis*	·Reduce sperm damage ·Improve sperm dynamics and morphology	[1]
pathogenic bacteria	*Pseudomonas aeruginosa*	Associated with defective spermatogenesis, sperm DNA damage, and orchitis	[2,3,4]
	*Escherichia coli*	·Associated with defective spermatogenesis, sperm DNA damage, and orchitis ·Affect sperm motility and morphology	[14,15,43,44]
	*Staphylococcus aureus*	·Associated with sperm DNA damage and orchitis ·Affects sperm viability and morphology	[5,6,7]
	*Prevotella*	Associated with defective spermatogenesis and low-quality semen	[8,9,10,11]
	*Brucella*	Orchitis	[12]
	*Chlamydia trachomatis*	·Associated with defective spermatogenesis, sperm DNA damage, and orchitis ·Affect sperm motility and morphology	[13]
	*Neisseria gonorrhoeae*	Associated with defective spermatogenesis, sperm DNA damage, and orchitis	[12]
	*Mycoplasma urealyticum*	·Associated with inflammation, sperm DNA damage, and orchitis ·Affects sperm viability and morphology	[12,13]
	*Staphylococcus saprophyticus*	Associated with poor sperm count, decreased sperm motility, abnormal viscosity, and leukocytospermia	[13]
	*Streptococcus agalactiae*		[13]
	*Klebsiella*		[1]
	*Bacillus citreus*		
	*Enterobacterium*		
	*Clostridium*		
	*Enterobacter cloacae*		
	*Aeromonas hydrophila*		

**Table 2 microorganisms-12-02253-t002:** Common natural antimicrobial substances in semen and their mechanisms of action.

Natural Antimicrobial Substances	Mechanisms of Action	References
Lysozyme (LSZ)	LSZ hydrolyzes the β-1,4 glycosidic bond between the NAM monomer and the adjacent NAG monomer. Hydrolysis of PG by lysozyme leads to cell wall instability and bacterial cell death. LSZ can also have a bactericidal effect through the mechanism of its cationic nature, i.e., the formation of pores in the negatively charged bacterial cell membranes by LSZ.	[63,64]
Secretory leukocyte peptidase inhibitor (SLPI)	Related to the special structure of the peptide chain, if the structure is changed, the antibacterial activity will decrease.	[65,66,67]
Lactoferrin (LF)	Inhibits and kills bacteria by highly binding iron, depriving them of the essential iron needed for growth.	[68,69]
Antibacterial peptides (AMPs)	The amphiphilic structure of AMPs, where the spatial separation of the cationic and hydrophobic components is a prerequisite for their effective interaction with bacterial membranes, is a structural feature that allows AMPs to interact with lipids of asymmetric bacterial membranes in a similar manner.	[60,70,71]
Group II phospholipase A2 (PLA2)	Catalyze the hydrolysis of phospholipids in the cell membrane of certain Gram-positive bacteria. Activates the body’s immune system and kills a variety of Gram-negative bacteria with the help of complement and other factors	[72,73]
Zn^2+^, SG, SGI-derived peptides and HEL-75 protein		[53]

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
