# Peer review of "The Equilibrium of Bacterial Microecosystem: Probiotics, Pathogenic Bacteria, and Natural Antimicrobial Substances in Semen"

_microorganisms, 2024, doi:10.3390/microorganisms12112253_

Round 1
Reviewer 1 Report
Comments and Suggestions for Authors
Predložený rukopis je zaujímavý a poskytuje prehľadné zhrnutie riešenej problematiky, mám však niekoľko formálnych pripomienok.
Skratky (LSZ, SLPI, LF, AMP) je potrebné definovať iba raz a potom použiť skratku namiesto celého názvu.
Pre E scherichia coli často používate skrátenú formu E. coli . Uvažovali by ste o použití tohto systému aj pre iné mikroorganizmy? Ak nie, použite prosím celý formulár aj pre E. coli .
V tabuľke 1 zjednoťte písmo a veľkosť Lactobacillus paracasei
Upevniť používanie kurzívy pre Lactobacillus, Bifidobacterium, Gonococcus
Pre názvy mikroorganizmov v riadkoch 161,195,198 použite kurzívu
Author Response
Dear reviewer:
Thank your comments, and we have replied your comment. We checked the whole manuscript, and corrected some spelling errors of words. Here is our response to your comments.
The manuscript presented is interesting and provides a clear summary of the problem, but I have a few formal comments.
Q1: Shortcuts (LSZ, SLPI, LF, AMP) need to be defined only once and then use the short instead of the full name
Response:Thanks for your valuable comments,we listed the full name of all abbreviations when the abbreviation first appeared. If it’s necessary to put a abbreviation list, we’ll add it in the manuscript.
Q2: For E scherichia coli you often use a short form of E. coli . Would you consider using this system also for other micro-organisms? If not, please use the full E. coli form as well.
Response:Thanks for you.We revised “E.coli” into “Escherichia coli”.
Q3: In Table 1, merge the letter and size of Lactobacillus paracasei
Response:Thanks for your valuable comments, the content of the corresponding place has been revised.
Q4: Strengthen use of the Lactobacillus, Bifidobacterium, Gonococcus in italics
Response:Thanks for your valuable comments, All bacterial name, these characters have been written in italics
Q5: For the names of micro-organisms in rows 161,195,198, use italics
Response:Thanks for you, the names of micro-organisms, these characters have been written in italics.

Reviewer 2 Report
Comments and Suggestions for Authors
Dear Authors,
This is an interesting review about the role of bacteria in semen. It is well organized, indicating the difference between bacteria communities and how they can affect the semen. However, for more clarification, you should distinguish between human semen and other animals' semen, is there any difference? You mentioned "mammalian semen", has this found in all mammals or are there any differences? Also, based on its applicability, I would include a section related to its implications on human reproduction and health, as well as another related to animal reproduction and health (i.e. in cattle).
Thank you so much.
Author Response
Reviewer 2
Dear reviewer:
Thank your comments, and we have replied your comment. We checked the whole manuscript, and corrected some spelling errors of words. Here is our response to your comments.
This is an interesting review about the role of bacteria in semen. It is well organized, indicating the difference between bacteria communities and how they can affect the semen.
Q1:However, for more clarification, you should distinguish between human semen and other animals' semen, is there any difference?
Response: Thanks for your valuable comments, in this review we only discuss other animals’semen and nothing related to humans semen. We have increased restriction, semen of common livestock
Q2:You mentioned "mammalian semen", has this found in all mammals or are there any differences?
Response:Thanks for your opinions. The types and predominance of bacteria and natural antimicrobial substances in semen are similar, although there are slight differences between species. The descriptions are marked in red in the text.
Q3:Also, based on its applicability, I would include a section related to its implications on human reproduction and health, as well as another related to animal reproduction and health (i.e. in cattle).
Response:Thanks for your valuable comments, the content on “Effects of semen on animal reproduction and health” has been added to the Applicability and prospects section (final two paragraphs of the text). The descriptions are marked in red in the text.

Round 2
Reviewer 2 Report
Comments and Suggestions for Authors
Hello Authors,
Thank you for your clarifications and addressing my comments.
Best wishes.